

# *In vivo* biotoxicological assessment of nanoplastics and microplastics predicted using the zebrafish model

Tao Ren[1,*], Libo Yan[2,*], Daogang Wang[3], Ning Xu[1], Weiming Zhang[4] and Mengzhe Yang[5]

[1] The Fifth Affiliated Hospital of Guangxi Medical University, Nanning, China
[2] Hubei University of Science and Technology, Xianning, China
[3] The First Affiliated Hospital of Guangxi University of Chinese Medicine, Nanning, China
[4] Wuming Hospital of Guangxi Medical University, Nanning, China
[5] Beijing Friendship Hospital, Capital Medical University, Beijing, China
[*] These authors contributed equally to this work.

## ABSTRACT

Nanoplastics (NPs) and microplastics (MPs) are emerging environmental pollutants that have raised concerns due to their potential impacts on human health. Zebrafish (Danio rerio) have been widely used as a model organism to study the toxicity of NPs and MPs and to evaluate the effects of these pollutants on human health. This review summarizes recent studies on the toxicities and potential effects of NPs and MPs in zebrafish and discusses how findings from this model can help predict their impact on human health. Additionally, the mechanisms by which NPs and MPs affect biological processes, such as growth, development, behavior, immune function, reproduction, and the nervous system, in zebrafish are further illustrated. Taken together, zebrafish serve as a valuable model for predicting the potential effects of NPs and MPs on human health and highlight the growing concern surrounding these environmental pollutants.

# INTRODUCTION

Nanoplastics (NPs) and microplastics (MPs) are small plastic particles that have become ubiquitous in the environment due to their widespread use in modern consumer products and industrial processes (*Cox et al., 2019*). These particles can enter the environmental ecosystem through various routes, including wastewater discharges (*Bayo, Olmos & López-Castellanos, 2020*), stormwater runoff (*Mason et al., 2016*), and atmospheric deposition (*Allen et al., 2020*). The potential impact of NPs and MPs on human and animal health has raised concerns in recent years (*Vethaak & Legler, 2021*; *Huang et al., 2021*). Once in the aquatic environment, NPs and MPs can be ingested by aquatic organisms, including fish, and can accumulate in their tissues and organs. Through the food chain, these particles can act as carriers of toxic chemicals, such as persistent organic pollutants (*Van Emmerik et al., 2018*) and heavy metals (*Qiao et al., 2019b*), and can cause physical damage to tissues and organs of biological organism including human beings (*Meaza, Toyoda & Wise Sr, 2020*).

Corresponding authors
Weiming Zhang,
weiming8207@163.com
Mengzhe Yang, yangmz10@163.com

However, the mechanisms underlying the toxicity of NPs and MPs are not well understood, and the effects of these particles on human health are largely unknown. In the present literature, the potential toxicity or side effect of nanoplastic and microplastic pollution were illustrated from the zebrafish model as an essential model organism on human health, which might highlight the concern on such kind of environmental pollutants in future.

The accumulation of plastic waste has become a global crisis (*MacLeod et al., 2021*), affecting our oceans, wildlife, and even the air (*Chae & An, 2017*). The problem stems from the fact that plastics are non-biodegradable, meaning they do not break down naturally like organic matter does (*Dobaradaran et al., 2018*; *Xiang et al., 2022*). Instead, it takes hundreds of years for them to decompose, during which time they can cause significant harm to the environment (*Auta, Emenike & Fauziah, 2017*). The sources of plastic pollution are varied and include everything from single-use packaging to discarded fishing nets (*Geyer, Jambeck & Law, 2017*). Plastic particles also degrade into smaller fragments, known as microplastics or even nanoplastics, which can then be ingested by marine life and eventually end up back in our food chain (*Zhou et al., 2020*). It was estimated that 75% of marine litter contained plastics (*Napper & Thompson, 2020*). Coastal countries generated more than 70,000 tonnes of plastic waste that entered the oceans in 2010 (*Jambeck et al., 2015*), and the accumulation of plastic particles ranges could not be despised (*Van Sebille et al., 2015*).

NPs and MPs are two types of plastic particles that pose a major environmental problem (*Sharma et al., 2023*; *Thompson et al., 2004*). *Thompson et al. (2004)* proposed microplastics in 2004 to quantify the abundance of microplastic in marine environment. NPs and MPs (large plastics breaking down into small pieces) are defined as nano and micro meter level (*Hartmann et al., 2019*). Both types of plastics can originate from a range of sources (*Rolsky et al., 2020*; *Saliba, Frantzi & Van Beukering, 2022*), including larger plastic debris that breaks down into smaller pieces over time, as well as products such as cosmetics and textiles which contain tiny plastic particles (*Rochman, 2018*; *Carney Almroth et al., 2018*). These particles can then accumulate in the environment, particularly in marine environments where they are ingested by wildlife, and can have negative impacts on their health (*Santos, Machovsky-Capuska & Andrades, 2021*).

In the environment, NPs and MPs can accumulate and disrupt ecosystems. They can be ingested by marine life, leading to physical harm or even death (*Hipfner et al., 2018*; *Kahane-Rapport et al., 2022*). Additionally, they could transport harmful chemicals into the food chain as plastics absorb toxic chemicals (*Song et al., 2022*) such as phthalates and Bisphenol A from surrounding seawater and sediments (*Danopoulos et al., 2020*). For human health, studies have shown that NPs and MPs can also enter the body through inhalation (*Chen, Feng & Wang, 2020*), ingestion (*Shruti et al., 2021*), and *via* skin contact (*Schirinzi et al., 2017*). However, the full impact on human health has yet to be thoroughly understood and is still an area of active research.

## The intended audience

The primary intended audiences for this review focus researchers in the fields of environmental toxicology and/or biology using zebrafish as research models. It provides a comprehensive summary of *in vivo* biotoxicological assessment of nanoplastic and

microplastic predicted by the zebrafish model, highlighting new perspectives on understanding the pathogenesis and relevant diseases associated with nanoplastic and microplastic pollutants.

## SURVEY METHODOLOGY

We conducted a comprehensive literature search across PubMed, Google Scholar, and Web of Science using keywords 'nanoplastic' OR 'microplastic' and a total of 3,663 and 18,798 articles published from January 2002 to December 2024 were hit, respectively. An additional keyword 'zebrafish' was included using AND to optimize the retrieval of relevant articles, and a total of 185 and 463 articles published from January 2002 to December 2024 were selected for inclusion in this review, respectively. The above terms were searched for in all parts of the article and the same article was recorded once. The types of research, meta-analysis, editorials and review articles in the English language were included. Letters and case reports were not included. The survey results were summarized and the representative MP and NP studies using zebrafish models were listed in Table 1, which indicated the different types of MP and NP and their effect on the zebrafish model.

### Zebrafish application in scientific research

The zebrafish (*Danio rerio*) has become a popular vertebrate model organism in scientific research due to its numerous advantages, including high fecundity, external fertilization, and transparent embryos (*Kimmel et al., 1995*) that allow for easy visualization of developmental processes. Research utilizing the zebrafish as a model organism is particularly prominent in the fields of genetics (*Muto et al., 2005*), developmental biology (*Yang et al., 2019*), neurobiology (*Niemeyer et al., 2022*), toxicology (*MacRae & Peterson, 2023*), cancer research (*White, Rose & Zon, 2013*), and drug discovery (*Patton, Zon & Langenau, 2021*).

In genetics, the rapid generation time and genome sequence similarity between zebrafish and humans make it an excellent organism to study genetic mutations that cause human diseases such as cancer (*Cagan, Zon & White, 2019*), muscular dystrophy and heart disease (*Zhao et al., 2021a*). In developmental biology, the transparency of the zebrafish embryo (*Lieschke & Currie, 2007*), provides a wealth of opportunities to study early embryonic development and organogenesis which allows researchers to visualize cellular processes in real-time at their earliest stages (*Batel et al., 2018*). Zebrafish also provide excellent models for neurological studies (*Wheeler et al., 2019*), with developing spinal cord and brain being easily monitored in transparent embryos. In addition, the ease of genetic manipulations makes this organism attractive to scientists researching psychiatric disorders (*Thyme et al., 2019*), personalized medicine (*García-López, Vilos & Feijóo, 2019*) and substance abuse (*Carmack et al., 2022*). Furthermore, because of its sensitivity to toxins and pollutants, the zebrafish is extensively used as a test organism in environmental toxicology (*McGrath & Li, 2008*). Studies using the zebrafish to examine how chemical exposure interrupts normal biological pathways and affects health have produced valuable information on harmful substances' effects on human beings. Finally, through high-throughput screening technology, analysis of complex zebrafish behaviors has been utilized for identifying drugs targeting specific diseases. Zebrafish not only serve as "disease simulators" but

**Table 1  Summary of representative microplastic (MP) and nanoplastic (NP) studies using zebrafish models.**

| Studies | Size of MPs/NPs | Accumulations | Effects |
|---|---|---|---|
| Brun et al., 2018 | 25 nm | gut, skin | aberrant gene expression |
| Brun et al., 2019 | 25 nm | gut, pancreas, gall bladder | malformations: swim bladder; higher cortisol; lower glucose |
| Pitt et al., 2018a | 40 nm | yolk, brain, heart, gut, pancreas, liver, gall bladder, chorion | decreased heart rate - hypoactive behavior in dark periods - locomotion: interaction of NP treatment and cohort in dark and light periods |
| Hu et al., 2020 | 40 nm | liver, gut, pancreas and gall bladder | carbohydrate metabolism, cell membrane biogenesis, immunity, endocytosis, and catalytic activity |
| Trevisan et al., 2019 | 44 nm | na | decreased mitochondrial coupling efficiency |
| Trevisan, Uzochukwu & Di Giulio, 2020 | 44 nm | na | slight decrease in mitochondrial coupling efficiency |
| Chen et al., 2017 | 50 nm | na | decreased body length, Reduced locomotor activity |
| Lee et al., 2019 | 50/200/500 nm | yolk, brain, retina, blood vessels, muscle, fascicles, spinal cord, CNS cells, chorion | na |
| Zhao et al., 2020a | 65 nm | gut, pancreas | delayed hatching, changed metabolites |
| Duan et al., 2020 | 100 nm | brain, gills, blood, liver, gut | metabolome affected |
| Liu et al., 2021 | 100 nm | na | transcriptome affected |
| Parenti et al., 2019 | 500 nm | na | transcriptome affected |
| Zhang et al., 2020a | 20/25/50/500 nm | chorion, yolk, eye, brain, and gut | na |
| Van Pomeren et al., 2017 | 50/25/700 nm | gill, skin, gut | na |
| Liu et al., 2019 | 50/100 nm | na | affected ROS level |
| Pedersen et al., 2020 | 200 nm | yolk, gut, pancreas, liver, ocular and cranial regions | decreased survival and hatching rate, - Increased malformations |
| Sökmen et al., 2020 | 20 nm | yolk and brain | increase in mortality |
| Brun et al., 2018 | 25 nm | na | na |
| Zhang et al., 2020b | 40 nm | eyes, head, yolk, gut | transcriptome affected |
| Veneman et al., 2017 | 700 nm | yolk, blood stream and heart | transcriptome affected |
| Pitt et al., 2018b | 40 nm | yolk, gut, liver, pancreas, gall bladder | decreased heart rate |

**Table 1** (*continued*)

| Studies | Size of MPs/NPs | Accumulations | Effects |
|---|---|---|---|
| *Lee et al., 2019* | 50/200/500 nm | chorion, eggs | exacerbated development abnormalities, survival, hatching rate along with increased production of ROS |
| *Qiang & Cheng, 2019* | 1 μm | chorion | decrease in swimming competence |
| *Wan et al., 2019* | 5–50 μm | intestinal tract | alterations in the microbiome |
| *Karami et al., 2017* | −17.6 μm | yolk, intestinal tract | na |
| *LeMoine et al., 2018* | 10–45 μm | intestinal tract | transcriptome affected |
| *Sleight et al., 2017* | 200–250 μm | na | reduced bioavailability of Phe and EE2 |
| *Sarasamma et al., 2020* | 70 nm | gonad, intestine, liver, and brain | disturbance of lipid and energy metabolism |
| *Lu et al., 2016* | 70 nm, 20 μm | gill, liver, gut | inflammation |
| *Gu et al., 2020* | 100 nm, 200 μm | intestine | dysfunction of intestinal immune cells |
| *Lei et al., 2018* | 1/50/70 μm | na | intestinal damage |
| *Qiao et al., 2019a* | 15/25 μm | gut | Intestinal mucosal damage |
| *Qiao et al., 2019b* | 10 nm, 20 μm | gut, liver, gill | increased level of malonaldehyde |
| *Jin et al., 2018* | 0.5/50 μm | na | transcriptome affected |
| *Batel et al., 2016* | 1–5, 10–20 μm | intestine | transcriptome affected |
| *Zhao et al., 2020b* | 5 μm | na | decrease in body weight |
| *Lu et al., 2018* | 6 μm | na | inflammation |
| *Qiao et al., 2019c* | 5 μm | gut, liver, gill | alterations in the microbiome |
| *Mak, Yeung & Chan, 2019* | 10–600 μm | intestine, gill | morphological changes |
| *Limonta et al., 2019* | 25/50/90 μm | na | transcriptome affected |
| *Rainieri et al., 2018* | 125–250 μm | na | transcriptome affected |
| *Kim et al., 2019* | 247.5 μm | na | recognize that MPs are not food items |
| *Batel et al., 2018* | 1–5, 10–20 μm | gills, intestinal tract, chorion | transcriptome affected |

also accelerate drug discovery by providing efficient tools for preclinical development (*Patton, Zon & Langenau, 2021*). Overall, the zebrafish's benefits make it an essential model organism in medical research.

Zebrafish also have become an important model organism in toxicology research due to their small size, high fecundity, and ease of maintenance as mentioned above (*Kimmel, 1989*). They possess a vertebrate physiology that is highly conserved with humans (*Howe et al., 2013*), making them a useful tool for studying the potential toxicity of various chemicals including NPs and MPs. The diameter of NP was between 20–200 nm and over 200 nm size was considered as MP. Zebrafish models including fertilized egg, embryo, larvae and adult were often used for these studies of NPs and MPs (Fig. 1). Recent advances in zebrafish research techniques, such as transgenic and gene-editing technologies (*Parvez et al., 2021*; *Hwang et al., 2013*; *Tran et al., 2007*), have further increased their usability in toxicological

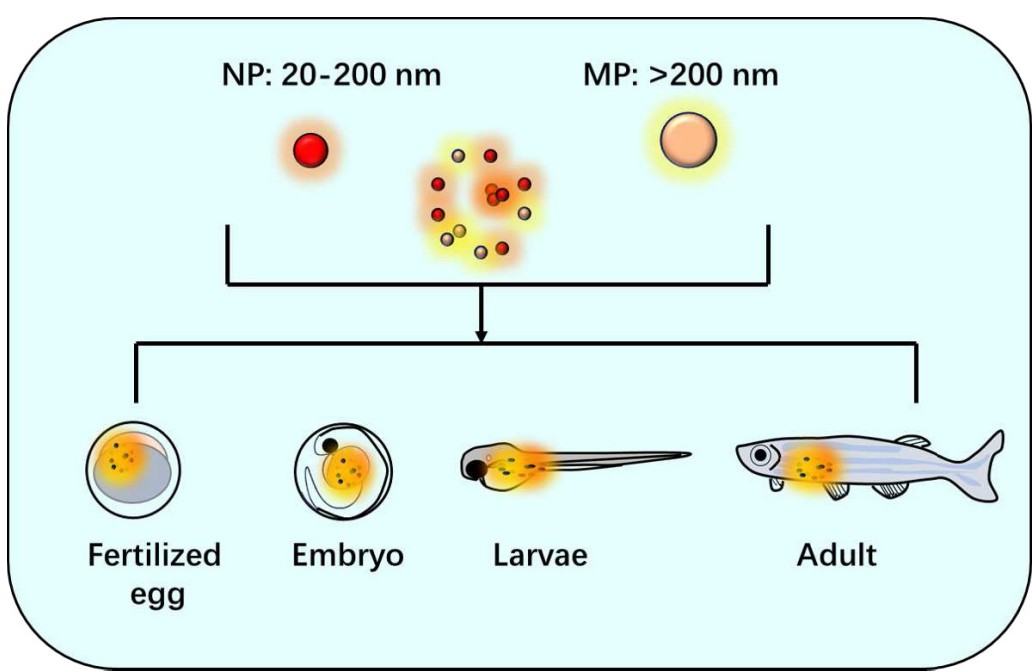

**Figure 1** Zebrafish as a research model of nanoplastic (NP) and microplastic (MP) studies.

studies. Zebrafish embryos are particularly valuable for screening chemical compounds for developmental toxicity and teratogenic effects (*Gustafson et al., 2012*). In addition, zebrafish models have been used to study the toxic effects of environmental pollutants, including pesticides, heavy metals, and microplastics (*Mak, Yeung & Chan, 2019*). Their transparent larvae and ability to genetically modify specific organs or physiological processes make them ideal for assessing the toxicity of different types of pollutants on specific organ systems or biological pathways. Overall, zebrafish provide a cost-effective and efficient alternative to traditional animal models for toxicological research while still allowing researchers to study complex physiological responses to environmental toxins.

## Zebrafish as an organism model in plastic toxicity studies

Current research on MPs and NPs in model organisms has predominantly utilized on *C. elegans* as a principal experimental system. *C. elegans* small size, transparency, stereotyped behaviors, cell linage, environmental manipulability, coupled with its expedited life cycle (3–4 days) and relatively brief life span (3 weeks) represent a great model organism for *in vivo* toxicology assessments (*Hoffschröer et al., 2021*). While mammalian models such as mice and rats remain crucial for translational research, their application in plastic particle studies faces substantial limitations due to ethical constraints imposed by institutional guidelines and complex husbandry requirements. In addition, emerging ecotoxicological evidence reveals the translocation and distribution of MPs in thousands of organisms, particularly aquatic animals (*Ma et al., 2021*). The zebrafish model organism model involves the studies to explore the effects of environmental pollutants on human health. In recent years, zebrafish has emerged as a powerful model organism for studying the

toxicity (*Qiao et al., 2019b*) and effects of plastics including NPs (*Torres-Ruiz et al., 2021*) and MPs (*Lu et al., 2016*). In addition, zebrafish share many physiological and genetic similarities with humans (*Howe et al., 2013*), making them a promising model organism for studying human health including the potential toxicity and effects of NPs and MPs. NPs and MPs can affect zebrafish growth, development, behavior, immune system, reproductive system, and nervous system (*Rojoni et al., 2024*). For example, *Qiao et al. (2019a)* and *Qiao et al. (2019c)* found that the accumulation of MPs in zebrafish caused multiple toxic effects and gut has an inflammatory and oxidative stress response after MPs exposure. The toxicity of these particles can be influenced by various factors, including particle size, shape, surface chemistry, and concentration. In addition, the toxic effects of NPs and MPs can be influenced by the developmental stage of zebrafish, with early life stages being more vulnerable to these particles than adult stages. Several studies have shown that NPs and MPs can be ingested by zebrafish and can accumulate in their tissues and organs (*Sarasamma et al., 2020*; *Araújo et al., 2022*). These particles can cause physical damage to tissues and organs, disrupt cellular processes, and induce oxidative stress and inflammation (*Teng et al., 2022b*). In addition, NPs and MPs can affect zebrafish behavior, including swimming activity, shoaling behavior, and predator avoidance behavior (*Sarasamma et al., 2020*; *Mattsson et al., 2017*). Moreover, these particles have been reported to affect zebrafish immune system, including the production of cytokines and the activity of immune cells. Till present, various methods have been developed to study the toxicity of NPs and MPs in zebrafish, including exposure experiments (*Zhao et al., 2020a*), behavioral assays (*Lu et al., 2016*), gene expression analysis (*Qiao et al., 2019b*). Behavioral assays involve measuring changes in zebrafish behavior in response to plastic exposure, using techniques such as video tracking and shoaling behavior analysis (*Jeong et al., 2022*). Gene expression analysis involves measuring changes in gene expression in zebrafish in response to plastic exposure, using techniques such as quantitative polymerase chain reaction (PCR) and RNA sequencing (*Gu et al., 2020*; *Veneman et al., 2017*). Histological analysis involves examining changes in tissue and organ morphology in zebrafish in response to NPs and MPs, using techniques such as light microscopy and electron microscopy (*Gu et al., 2020*). This model has several advantages over traditional *in vitro* and *in vivo* models, including the ability to study the effects of environmental pollutants on human cells and tissues in a living organism, the ability to evaluate the systemic effects of environmental pollutants on human health, and the ability to screen for potential therapeutic agents. Overall, zebrafish is a promising model organism for studying the toxicity and effects of NPs and MPs. The zebrafish model organism model has the potential to revolutionize our understanding of the impact of environmental pollutants particularly NPs and MPs on human health.

There are several studies on the effects of NPs and MPs in zebrafish. One recent study found that exposing zebrafish embryos to different particle sizes polystyrene nanoplastics led to changes in gene expression related to inflammation and cardiac development, delayed embryo hatching (*Zhou et al., 2023*). Another study showed that exposing adult zebrafish to microplastics (1–5 μm) resulted in altered gut microbiota composition and increased oxidative stress (*Qiao et al., 2019c*). Overall, these studies demonstrate the potential negative effects of NPs and MPs, which could have broader implications for aquatic ecosystems and

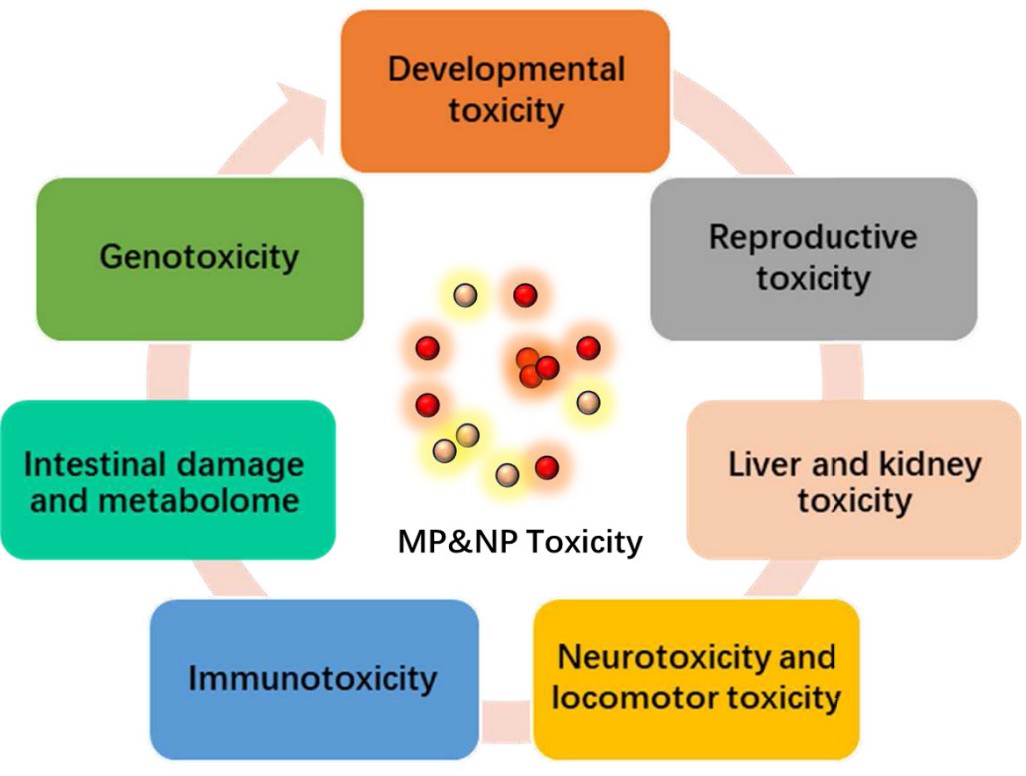

**Figure 2** The relevant biological toxicity effects caused by nanoplastics (NPs) and microplastics (MPs).

human health if similar effects occur in other animal species at higher levels of exposure or reacting with other contaminants. The NPs, MPs often mixed and reacted with some other contaminants, which induced their changes of shapes, properties and affected the sorption or desorption and the subsequent aggregation/accumulation or transformation/speciation. The caused toxicity displayed different biological effects (Fig. 2). The relevant information was described as follows:

### Uptake and accumulation of NPs and MPs

There have been several studies conducted to investigate the uptake and accumulation of NPs and MPs using zebrafish models. One study, for example, found that exposure to polystyrene nanoparticles caused an increased accumulation of nanoparticles in the liver and brain of adult zebrafish, as well as decreased overall movement and increased anxiety-like behavior (*Torres-Ruiz et al., 2023*). Another study found that embryonic and larval zebrafish exposed to microplastics had higher mortality rates, slower growth, and altered gene expression patterns compared to control groups (*Duan et al., 2020*). Additionally, plastic particles were observed in the gastrointestinal tract, gills, and other tissues of the fish (*Qiao et al., 2019b*).

### Impact of different types/size/shapes of plastics

Research has shown that different types, sizes and shapes of plastic can have varying impacts on zebrafish. For example, a study found that exposure to microbeads made of polyethylene caused oxidative stress in the livers of zebrafish, while polystyrene particles had no significant effect (*Lu et al., 2016*). The researchers also found that smaller particle size resulted in higher levels of ingestion by the fish (*Jeong et al., 2017*). The particle size of MPs determines their uptake and distribution in organisms. *Lee et al. (2019)* assayed three different sizes (diameters of 50, 200 and 500 nm) of polystyrene nanoplastics in zebrafish embryo, and observed that 50 nm particles accumulated in the chorion and embryo; whereas 200 nm and 500 nm particles accumulated in the chorioallantoic membrane, and very weakly in the zebrafish embryo. Another study investigated the impact of differently shaped particles on zebrafish embryos (*Bhagat et al., 2020*). They found that exposure to spherical polystyrene nanoparticles led to developmental abnormalities, while exposure to rod-shaped particles resulted in decreased hatching rates but no observable developmental abnormalities. Additionally, studies have highlighted the importance of considering the combination of plastics as well as their interactions with other pollutants in the environment. One study found that co-exposure to nanoplastics and pesticides resulted in increased neurotoxicity in zebrafish larvae (*Bhagat et al., 2021*). Overall, these studies demonstrate that the properties of plastics such as type, size, and shape can have important implications for their toxicity to aquatic organisms like zebrafish.

### NPs and MPs as carriers for other pollutants

NPs and MPs can act as carriers for other pollutants, which can further increase their potential accumulation effects. As studies on these particles deepen, researchers have found that they tend to aggregate into composites in order to adsorb toxins present in the environment, such as heavy metals, pesticides, and persistent organic pollutants (*Lee et al., 2019*; *Koelmans, Besseling & Foekema, 2014*). NPs are more easily taken up by organisms and transferred to higher-level consumers in the food chain. When pollutants combine with nano- or microplastics, it can lead to even greater potential accumulation effects. For example, in one experiment, zebrafish were exposed to wastewater containing two types of nanoplastic, along with diphenyl oxide calcium and naphthylamine; after two weeks, both types of plastic were found in the fish tissue (*Bhagat et al., 2020*). These findings highlight the need for further study on the interaction between plastics and other environmental toxins, as well as how this interaction can impact ecological systems and human health. Effective measures should be implemented to prevent plastic pollution from escalating and protect entire ecosystems.

In recent years, the accumulation of NPs and MPs in aquatic environments has become a major environmental concern. These small plastic particles have been found to impact different organisms in various ways including zebrafish. Several studies have investigated the toxicities of NPs and MPs on zebrafish. One study showed that exposure to polystyrene nanoplastics reduced the hatching rate and increased the mortality rate in zebrafish embryos (*Bashirova et al., 2023*). It has also been observed that microplastics can cause changes in gene expression and oxidative stress in adult zebrafish (*Mak, Yeung & Chan, 2019*; *Kim et*

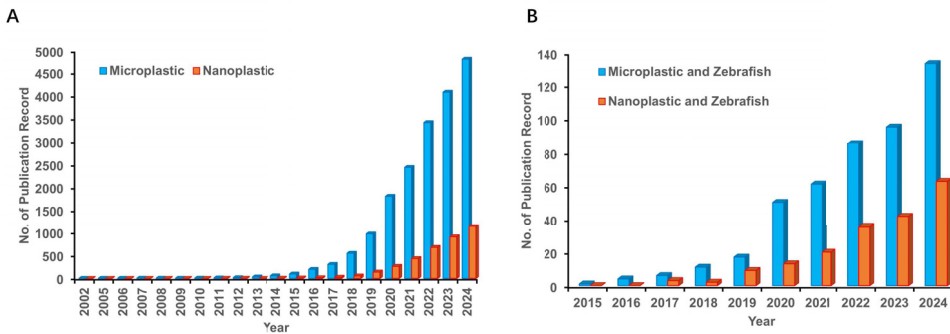

**Figure 3** The rapid developed publication records about nanoplastic (NP) and microplastic (MP) (A) and relevant research using zebrafish models (B).

*al., 2021*). Another study reported that exposure to polyethylene microplastics caused lipid peroxidation and DNA damage in zebrafish cells (*Wang et al., 2022*). Moreover, ingestion of microplastics by zooplankton which are important components of the zebrafish diet has led to behavioral changes such as reduced feeding rates, slower swimming speeds, and impaired predator avoidance responses (*Qiang & Cheng, 2019*; *Chen et al., 2020*; *Wright et al., 2013*). This suggests that these plastics may have multiple toxic effects during the embryonic stage and adult stage. The main toxicity includes developmental toxicity, reproductive toxicity, neurotoxicity and locomotor toxicity, immunotoxicity, genotoxicity, liver and kidney toxicity, intestinal damage and metabolome (Fig. 3). The representative studies were summarized in Table 1, and some relevant detailed information were illustrated as follows:

### Developmental toxicity

NPs and MPs have been found to cause developmental toxicity in zebrafish, a commonly used model organism for environmental toxicology studies. The results showed that both NPs and MPs caused dose-dependent developmental abnormalities in the zebrafish embryos, including delayed hatching, yolk sac malabsorption, spinal deformities, and reduced body length (*Batel et al., 2018*). Another study investigated the effects of polystyrene microplastics exert on zebrafish heart, including fish activity, metabolic changes and oxidative stress (*Wang et al., 2022*). The researchers observed that exposure to MPs led to decreased heart rate, abnormal cardiac morphology and a significant decrease in swimming velocity (*Dimitriadi et al., 2021*). The mechanisms underlying the developmental toxicity of NPs and MPs in zebrafish are not completely understood, but several hypotheses have been proposed. One theory is that these small plastic particles can disrupt the normal functioning of cellular processes such as gene expression and oxidative stress response, induced developmental toxicity with microcirculation dysfunction, leading to developmental abnormalities (*Park & Kim, 2022*). Another hypothesis suggests that NPs and MPs can adsorb and accumulate other environmental toxins (*Xu et al., 2021*), such as heavy metals (*Qiao et al., 2019b*) and organic compounds, which can further exacerbate their toxicity.

### Reproductive toxicity

There is increasing concern about their potential to cause adverse effects on aquatic organisms, including reproductive toxicity. Research has shown that exposure to NPs and MPs can affect the reproductive physiology. For example, one study found that male zebrafish exposed to microplastics had decreased sperm quality and altered gene expression related to reproductive hormone signaling (*Qiang & Cheng, 2021*). Another study showed that female zebrafish exposed to nano-sized polystyrene particles had reduced egg laying capacity and altered transcriptomic profiles related to ovarian development (*Zuo et al., 2021*). Disruption of the oogenesis process was measured by upregulation of vitellogenin (vtg1) in MP-exposed zebrafish (*Mak, Yeung & Chan, 2019*). Moreover, polystyrene microplastic increased the accumulation of microcystin-LR, produced by cyanobacterial species, in the gonads of zebrafish and enhanced reproductive disruption (*Lin et al., 2023*). These adverse effects on reproduction may be due, in part, to the ability of these particles to interact with endocrine disrupting chemicals (EDCs) and other contaminants that can also be present in the aquatic environment (*He, Yang & Liu, 2021*). NPs and MPs can act as carriers for these compounds, facilitating their uptake by aquatic organisms and potentially enhancing their toxic effects (*Yang et al., 2022*). Furthermore, these plastic particles have been shown to induce oxidative stress and inflammation in fish, which can lead to cellular damage and dysfunction (*Santos et al., 2020*). This can ultimately impact an organism's overall health and contribute to reproductive abnormalities.

Overall, the available scientific evidence suggests that NPs and MPs can pose a risk to the reproductive health of aquatic organisms, including fish like zebrafish. More research is needed to fully understand the mechanisms underlying these effects and to develop effective strategies for mitigating their impacts.

### Neurotoxicity and locomotor toxicity

Studies have shown that these small plastic particles can have negative impacts on aquatic organisms, including neurotoxic effects on zebrafish. For example, one study found that exposure to nanoplastics led to changes in the brain development of zebrafish embryos, resulting in morphological abnormalities and behavioral changes (*Mak, Yeung & Chan, 2019*; *Teng et al., 2022a*). Another study demonstrated that exposure to microplastics impairs spatial learning and memory consolidation in adult zebrafish (*Yu et al., 2022*; *Santos et al., 2021*). In this study, fish exposed to microplastics had difficulty navigating through a maze and exhibited reduced activity levels compared to control groups. *Chen et al. (2017)* found that nanoplastics are significantly more developmental neurotoxic to zebrafish larvae than microplastics. Exposure of zebrafish larvae to these plastic particles affects motor behaviour and may pose a risk to aquatic organisms. These findings suggest that NPs and MPs can have significant impacts on the neurological function of aquatic organisms, potentially leading to decreased fitness and survival rates.

### Immunotoxicity

NPs and MPs are two types of plastic particles that have been found to have negative impacts on the immune system of zebrafish. Studies have shown that exposure to nanoplastics can disrupt the innate immune response of zebrafish, which is responsible for protecting the

organism from infections. Polystyrene microplastics can induce hepatic inflammation in zebrafish larvae, affecting on neutrophils and macrophages (*Dimitriadi et al., 2021*; *Cheng et al., 2022*). One study found that zebrafish embryos exposed to nanoplastics exhibited abnormal development of their brains and had a higher mortality rate compared to control groups that were not exposed to nanoplastics (*Zuo et al., 2021*; *Sökmen et al., 2020*). Even after a short exposure, nanoplastics can still infiltrate zebrafish embryo tissues (*Parenti et al., 2019*). Additionally, researchers found that nanoparticles could accumulate in the gills of adult zebrafish, impairing their respiratory function and making them more susceptible to infections (*Wan et al., 2019*). Similarly, microplastics have also been linked to reduced immunity in zebrafish. In one study, adult zebrafish exposed to microplastics showed altered spatial learning and memory and had impaired survival rates (*Limonta et al., 2019*). Researchers hypothesize that this may be due to the ability of microplastics to adsorb toxic chemicals and pathogens, leading to an accumulation of harmful substances in the fish's tissues. Combined single-cell RNA sequencing, *Gu et al. (2020)* revealed different sizes of NPs and MPs induced dysfunction of intestinal immune cells and increased the abundance of pathogenic bacteria.

### Genotoxicity

Many studies have investigated the potential toxic effects of these plastic particles on various organisms, also including zebrafish. A study found that exposure to nanoplastics can lead to abnormal brain development in zebrafish embryos. The researchers exposed the embryos to low concentrations of polystyrene nanoparticles and observed significant alterations in neural and behavioral development (*Barboza et al., 2020*). Another study examined the effects of microplastic exposure on adult zebrafish (*Batel et al., 2018*). The researchers found that chronic exposure to environmentally relevant concentrations of microplastics led to spatial learning and memory deficits. These cognitive impairments were associated with changes in gene expression related to neurobehavioral function. Overall, research suggests that both NPs and MPs can have harmful genetic and neurological impacts on zebrafish.

### Liver and kidney toxicity

In terms of liver toxicity, research indicates that exposure to NPs and MPs can lead to increased levels of oxidative stress in the liver tissue of zebrafish. This oxidative stress can cause damage to the liver cells, leading to inflammation and impairing liver function. Moreover, the accumulation of these plastics in the liver can affect the lipid metabolism of zebrafish, potentially causing lipid droplet formation and disrupting normal hepatocyte functions (*Zhao et al., 2020b*). As for kidney toxicity (*Pitt et al., 2018*), studies suggest that NPs and MPs can accumulate in the glomeruli and tubules of zebrafish kidneys, resulting in cellular injuries such as oxidative stress, inflammation, and apoptosis. This accumulation can further impede the excretion process in zebrafish kidneys, which affects their normal renal function. It is worth noting that the different types of plastics and their sizes can affect the degree of toxicity on zebrafish organs (*Qin et al., 2021*). The toxic effects may also vary depending on the duration and level of exposure to NPs and MPs.

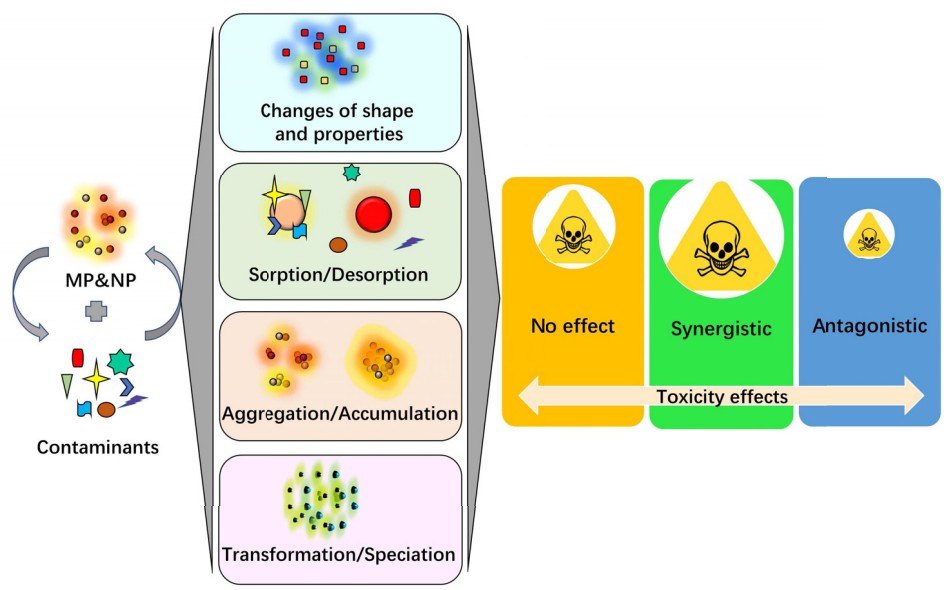

**Figure 4** The involved interactions of nanoplastic (NP) /microplastic (MP) with co-contaminants and affected toxicity effects.

### Intestinal damage and metabolome

Recent studies have indicated that exposure to these plastic particles can cause serious damage to fish, including intestinal damage and changes to the metabolome (*Wan et al., 2019*). A study on zebrafish has shown that exposure to nanoparticles resulted in significant damage to the intestinal lining. This damage was observed through histological analysis, which revealed extensive disruption of cell walls and loss of microvilli (*Lei et al., 2018*). Similarly, exposure to microplastics caused intestinal inflammation as well as changes to the structure of intestinal villi (*Jin et al., 2018*). At the same time, both NPs and MPs were found to alter the metabolome of zebrafish (*Zhao et al., 2021b*). Specifically, exposure to these plastic particles led to significant changes in the levels of certain amino acids, lipids, sugars, and energy-related metabolites. These changes suggest disruptions in metabolic pathways, which could lead to negative health outcomes.

## Zebrafish research about NPs and MPs to human health effects

*Huang et al. (2024)* established a generalized adverse outcome pathway (AOP) framework to delineate the developmental toxicity mechanisms of MPs and NPs, providing a systematic approach to predict the potential developmental toxicity of MPs and NPs to organisms and prioritize hazard identification for regulatory risk assessment. To advance human health impact evaluations, testing methods in *in vivo* quantification of plastic polymer bioaccumulation are critically required. It can establish a direct link between plastic exposure and potential health disorders. Furthermore, understanding of how NPs and MPs interact at the cellular and molecular levels, particularly particle-cell membrane interactions, intracellular trafficking mechanisms, and organelle-specific bioactivity, is

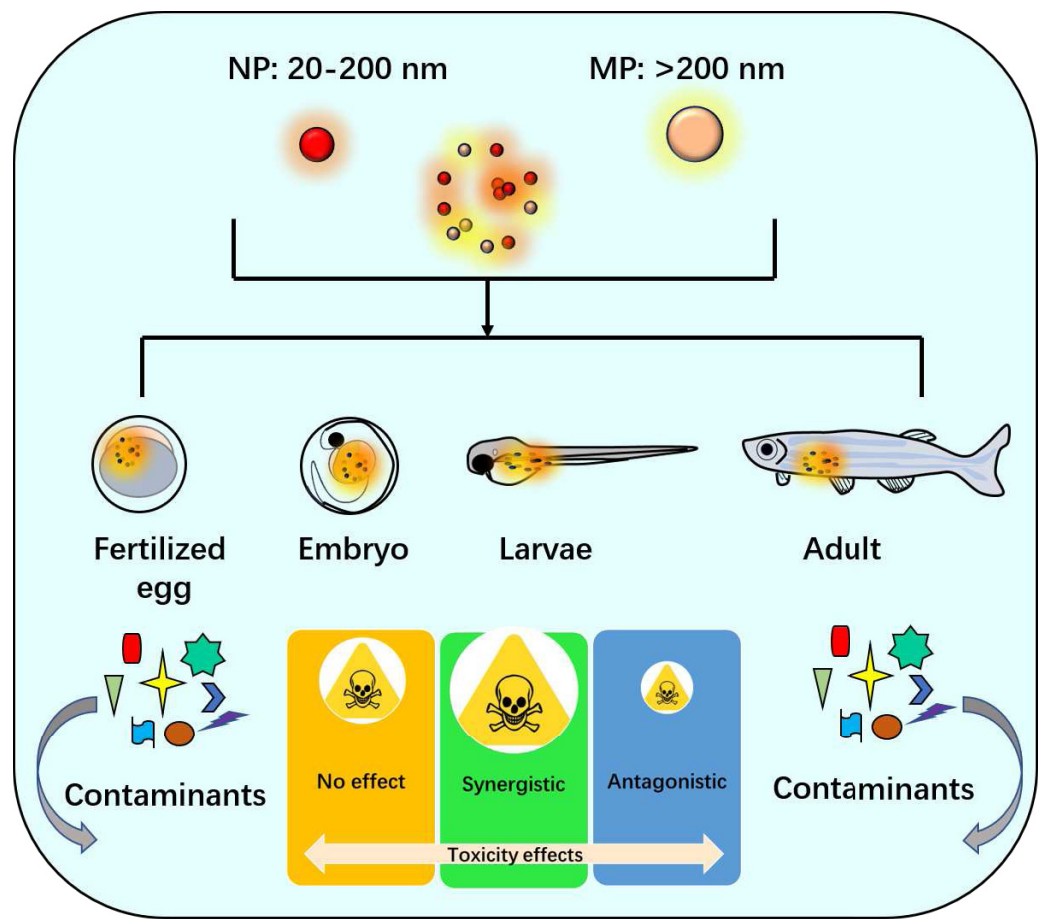

**Figure 5** A summarized map of nanoplastic (NP) and microplastic (MP) studies using different zebrafish models displayed various toxicity effects.

crucial to extrapolating the potential risks these materials pose to human health (*Winiarska, Jutel & Zemelka-Wiacek, 2024*).

## CONCLUSION AND PERSPECTIVE

Taken together, the needs were emphasized for further exploration of the mechanisms underlying the toxicities of NPs and MPs in zebrafish and the development of more accurate and reliable methods for evaluating their impact on human health, along with the fact that relevant studies have rapid developed (Fig. 4). The translation of zebrafish research about NPs and MPs to human health effects is still an open question, which requires further concerns. For example, the multi-omics approaches will be recommended to compare the effects of these MPs and NPs between different research models and human beings. In addition, the importance of interdisciplinary collaboration between researchers in different fields, including toxicology, ecology and medicine to address this complex and urgent issue were also highlighted. Studying nanoplastic and microplastic toxicity in zebrafish can be challenging due to various factors, including the complexity of the aquatic ecosystem,

the difference of animal models and human beings, the variability of nanoplastic and microplastic properties, and the limitations of current experimental techniques (Fig. 5). Anyhow, the translation of zebrafish research about NPs and MPs to human health effects is still an open question, which requires further concerns.

### Funding

This research was funded by the Project of Department of Science and Technology of Guangxi Zhuang Autonomous Region, China (Grant no. Guike AB19110052), Natural Science Foundation of Yunnan, China (grant number 202101AU070130), Key Research Project of Wuming, Nanning (Grant no. 20210124), Self-funded Research Project of Guangxi Medicine (Grant no. GXZYA20220270) and Program of Young Teacher and Researcher Development in Guangxi Province (Grant no. 2022KY0075). The funders had no role in study design, data collection and analysis, decision to publish, or preparation of the manuscript.

### Grant Disclosures

The following grant information was disclosed by the authors:
Project of Department of Science and Technology of Guangxi Zhuang Autonomous Region, China: Guike AB19110052.
Natural Science Foundation of Yunnan, China: 202101AU070130.
Key Research Project of Wuming, Nanning: 20210124.
Self-funded Research Project of Guangxi Medicine: GXZYA20220270.
Program of Young Teacher and Researcher Development in Guangxi Province: 2022KY0075.

### Competing Interests

The authors declare there are no competing interests.

### Author Contributions

- Tao Ren performed the experiments, analyzed the data, authored or reviewed drafts of the article, and approved the final draft.
- Libo Yan performed the experiments, analyzed the data, prepared figures and/or tables, and approved the final draft.
- Daogang Wang performed the experiments, prepared figures and/or tables, and approved the final draft.
- Ning Xu performed the experiments, prepared figures and/or tables, and approved the final draft.
- Weiming Zhang conceived and designed the experiments, performed the experiments, analyzed the data, authored or reviewed drafts of the article, and approved the final draft.

- Mengzhe Yang conceived and designed the experiments, performed the experiments, analyzed the data, authored or reviewed drafts of the article, and approved the final draft.

## Data Availability

This is a literature review.

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
