# Peer review of "In vivo biotoxicological assessment of nanoplastics and microplastics predicted using the zebrafish model"

_PeerJ, doi:10.7717/peerj.19833_

## Round 0.1 · original submission · Major Revisions

Reviewer 1 ·

Basic reporting

I have completed my review on the manuscript entitled “In vivo biotoxicological assessment of nanoplastic and microplastic predicted by the zebrafish model.” My comments are as follows:
1. As there are several reviews on the same topic, why do we need another review of NP and MP on zebrafish? What new information does this manuscript provide compared to other reviews in the same field?
2. For a review, the Methods and Materials section must be clear, and the criteria for article selection are crucial. Please show details on how the authors searched for and selected papers. Why are there no papers published in 2024 and 2025 included? Here are two for your reference:
uang, W., Mo, J., Li, J., & Wu, K. (2024). Exploring developmental toxicity of microplastics and nanoplastics (MNPS): Insights from investigations using zebrafish embryos. Science of the Total Environment, 173012.
Rojoni, S. A., Ahmed, M. T., Rahman, M., Hossain, M. M. M., Ali, M. S., & Haq, M. (2024). Advances of microplastics ingestion on the morphological and behavioral conditions of model zebrafish: A review. Aquatic Toxicology, 106977.

3. There is no synthesized information in the manuscript. The three figures and one table provided are too simple. Why are there no supplementary data provided? Have the authors read and interpreted the papers carefully?
4. It seems that no studies have been performed to investigate the molecular mechanisms underlying the toxicity induced by NP and MP. Please clarify.
5. What are the research gaps that need to be filled by future studies?
6. How can the studies from zebrafish be extrapolated to humans? Please provide more details.

Experimental design

improvement is needed

Validity of the findings

improvement is needed

Reviewer 2 ·

Basic reporting

Yes, the review of broad and cross-disciplinary interest and within the scope of the journal.
Yes, he field been reviewed recently, and yes there a good reason for this review (different point of view, accessible to a different audience, etc.).
Yesthe Introduction adequately introduce the subject and make it clear who the audience is/what the motivation is.

Experimental design

Yes, the Survey Methodology is consistent with a comprehensive, unbiased coverage of the subject.
Yes, the sources are adequately cited. But for the detailed informations on the zebrafish model and micro-and nano- plastics authors may refer and cite the mentioned article links which are mentioned in the additional comments section.
Yes, the review organized logically into coherent paragraphs/subsections.

Validity of the findings

Yes, there is a well-developed and supported argument that meets the goals set out in the Introduction.
The conclusion must include a future perspective and the future impact of micro-and nano- plastics on the environment and how can we reduce the use or negative impacts of them on health and environment.

Additional comments

Authors must report the details of the Zebrafish model with more clarity and also mention about the studies of MPs and NPs done till-date using some other model organisms (for example, the Mouse model, Drosophila model, etc.) used in the current scientific era and how the zebrafish model serves as the best model organism in comparison to others.
The figures used or mentioned by the authors must be revised with a detailed explanation and demonstration of the micro- and nano-plastic exposures to the zebrafish model.
In the section of "SURVEY METHODOLOGY" the Authors must also mention a table indicating the different types of micro- and nano-plastics and their effect on the Zebrafish model, along with the respective references.

---

## Round 0.2 · Minor Revisions

Thank you for revising the manuscript according to reviewers suggestions. I am glad to accept it conditional to the inclusion of few more details in the search strategies. Specifically, indicate if the terms were searched for in all parts of the article or in the title. And also specifiy if you use AND or OR, etc.

Reviewer 1 ·

Basic reporting

-

Experimental design

-

Validity of the findings

-

Additional comments

The manuscript was improved.

Reviewer 2 ·

Basic reporting

Yes, the review of broad and cross-disciplinary in interest and within the scope of the journal.

Experimental design

Yes, the Survey Methodology is consistent with a comprehensive, unbiased coverage of the subject.

Validity of the findings

Yes, there is a well-developed and supported argument that meets the goals set out in the Introduction.

---

## Round 0.3 · accepted · Accept

Thank you for providing more information in the revised manuscript. It is a pleasure to accept it.